# Revitalizing Rural Tourism: A Croatian Case Study in Sustainable Practices

**Marina Funduk** , **Ivana Biondić *** and **Abra Lea Simonić**

Institute for Development and International Relations, Lj. F. Vukotinovića 2, 10000 Zagreb, Croatia; marina@irmo.hr (M.F.); abraleas@gmail.com (A.L.S.)
* Correspondence: ibiondic@irmo.hr

**Abstract:** Dubrovnik-Neretva County, renowned for the City of Dubrovnik, grapples with tourism challenges affecting its UNESCO-listed Old City. This study advocates for promoting less-explored inland areas to ease the strain on the heritage site and alleviate coastal tourist pressure. By diversifying tourism and supporting sustainable rural development, the region can spur economic growth, foster local businesses, and improve infrastructure through EU and national funding. The research explores financial investments in less-developed areas, emphasizing sustainable tourism practices for socio-economic and environmental benefits. The analysis highlights the project's positive impact on sustainable tourism development inland, preserving natural heritage, and fostering economic benefits for local communities. According to the cost–benefit analysis, the proposed idea outperforms all alternatives with new attractions and enhanced infrastructure, contributing to overall municipal growth. External funding is crucial for viability, with a negative net income until 2040, offset by municipal support. Economic indicators justify social and economic benefits, emphasizing project resilience. The active tourism centre project, emphasizing eco-friendly outdoor activities, highlights the role of cost–benefit analysis in rural tourism infrastructure investment, recommending external funding for success.

**Keywords:** sustainable tourism; rural development; Dubrovnik-Neretva County; cost–benefit analysis; centre for active tourism

## 1. Introduction

The Dubrovnik-Neretva County, located at the southernmost tip of the Republic of Croatia, is renowned for the City of Dubrovnik. This world-famous city stands as one of Europe's most sought-after coastal destinations. The Old City of Dubrovnik has been designated as a UNESCO World Heritage site since 1979 and is renowned as Croatia's most visited tourist destination. Consequently, the areas surrounding Dubrovnik and its associated islands enjoy immense popularity among tourists. However, other parts of the county, mainly inland, experience a different level of tourism. One significant reason for promoting tourism in these less-developed areas of the county is the impact that the high number of tourists has on the heritage site of the Old Town. Recent surges in tourist numbers have raised pressing questions about its long-term preservation and conservation [1,2]. Overcrowding, environmental degradation and potential loss of authenticity have emerged as significant challenges facing the Old City of Dubrovnik. The strain that has been put on the city's infrastructure, historical sites and natural surroundings has started a debate about the need for more sustainable tourism practices [3]. Encouraging these practices can serve as a balance between tourism and environmental conservation, local culture, community engagement, employment opportunities and economic growth. Overtourism, as experienced in Dubrovnik, is characterised in the literature as a situation where the number of tourists on the streets negatively affects residents' quality of life [4] or, creates unsafe conditions or leaves no room for additional visitors [5,6].

Promoting the surrounding areas of Dubrovnik-Neretva County as attractive tourist destinations could divert some tourists from the Old City [1]. Since there are numerous various landscapes, historic towns and unique cultural traditions in Dubrovnik-Neretva County that still need to be explored by most tourists, expanding tourism beyond Dubrovnik could advance the region's economic growth. The interior of Dubrovnik-Neretva County has witnessed substantial emigration and a notable decline in population. Over two decades (2001 to 2021), the resident count diminished by 12%. This number is mitigated to some extent by larger towns such as Metković and Opuzen, but in some municipalities, the drop amounts to almost 40% [7]. Most of the population emigrates to Dubrovnik or other Croatian counties, while they emigrate abroad to a lesser extent. Furthermore, agriculture is the most developed of the economic branches, while tourism is less developed than in the rest of the county. According to the tourism development index calculated by the Institute for Tourism, Dubrovnik received a rating of 35.2, while the country's interior is 3.4 [8]. In other words, while the coast is in the first development group, the county's interior, which is up to 100 kilometres from Dubrovnik, is in the fourth development group. Other vital problems of Dubrovnik-Neretva County inland are identified in the Development Plan until 2027 and include problems like the ageing population, insufficient number of health workers and the number of preschool institutions that require reconstruction or construction [7]. The lack of environmental awareness and a coherent vision for environmental infrastructure decision-making further hinder progress. Outdated road infrastructure and inadequate regional connectivity compound the challenges, particularly affecting the inland areas of Dubrovnik-Neretva County, highlighting the pressing need for comprehensive planning and strategic initiatives to enhance sustainable tourism development. Expanding tourism inland may influence emergence of new businesses, restaurants, accommodations, and tour operators. These business ventures can increase local communities' revenue and stimulate economic development. Funding from the EU and national resources can foster infrastructure development, transportation, and additional accommodation, enhancing the visitor experience.

Therefore, this paper analyses financial investments in touristic infrastructure in less-developed areas in Dubrovnik-Neretva County inland. It also explores possibilities for the sustainable tourism development of those rural areas.

The main aim of this paper is to research the benefits of financial investments in the tourist revitalization of Dubrovnik-Neretva County inland areas and to promote the sustainable advancement of less-developed tourist destinations.

The hypotheses of this research are as follows:

**Hypothesis 1 (H1).** *Financial investments in Dubrovnik-Neretva inland areas encourage touristic development.*

**Hypothesis 2 (H2).** *Revitalization of less-developed areas requires external funding.*

The paper starts with a literature review and an explanation of the strategic documents in Croatia related to European Union (EU) funding. It continues with an explanation of the methodology, which includes a description of the cost–benefit analysis (CBA) method used in the research. The proposed project in Dubrovnik-Neretva county is the central part of the paper, with the final section dedicated to discussing the results of the conducted analysis and presenting its key findings.

## 2. Literature Review

### 2.1. Sustainable Tourism: Environmental Focus and Community Participation

When we delve into literature and the multifaceted role of sustainable tourism, it shows us that it is not merely a trend but represents a needed shift in how we perceive and practice tourism. It displays sustainable tourism as a growing type that benefits both the tourists and the destination in question [9,10]. It also defines it as a type of tourism where people pursue activities that benefit the countryside and natural heritage [11]. The original

idea of sustainable tourism emerged in the late 1980s [12], and it was considered to be a direct response to global concerns, such as environmental and cultural degradation, as a consequence of tourism expansion [13]. The United Nations World Tourism Organization defines sustainable tourism as tourism that yields current and future economic, environmental, and social impacts on the needs of the industry, visitors, the host community, and the environment [14]. In a broader context, sustainable tourism strives for economic growth, environmental preservation, and social engagement. Economic sustainability ensures that tourism initiatives generate income, improve livelihoods, and contribute to the further economic growth of local communities. The environmental aspects ensure that tourism practices will minimise the adverse effects on the ecosystem, cultural and natural heritage, and the overall well-being of the environment. Social sustainability revolves around community engagement, empowerment of local stakeholders and cultural heritage preservation [15,16]. Moreover, research on tourism sustainability has been increasing due to the contribution of tourism to economic development [17].

One of the key pillars of sustainable tourism is environmental preservation. There has been a global effort to grow more conscious of the impact we, as individuals, hold on our planet. Therefore, sustainable tourism promotes practices to reduce these adverse impacts, decrease carbon footprints, minimise waste, and conserve natural and cultural resources [18].

Environmental preservation through sustainable tourism is not a one-size-fits-all approach. Destinations are encouraged to adapt their initiatives to their ecological and cultural contexts. In ecologically sensitive areas, there is an emphasis on low-impact tourism practices, such as carrying capacity, that prevent overcrowding and further degradation [19]. One of the forms of sustainable tourism activities that helps conserve the natural environment is bird-watching. Promoting bird-watching allows tourists to appreciate nature, diverse wildlife, and unique landscapes. It engages with local communities, contributing to ecological conservation and regional economic development [20]. This approach aligns with low-impact tourism principles, minimising harm to sensitive habitats, but also serves as an economic catalyst for less-developed regions by attracting visitors around the globe, generating revenue, and creating job opportunities. Activities such as relaxation, rejuvenation, education, working on surrounding forests and bird-watching fall under the category of eco-tourism, which is a form of tourism that aims to appreciate nature, and it typically includes activities that allow tourists to engage with the natural world while minimising negative impacts on the environment [21]. It aims to promote conservation and sustainability, educate tourists on the environment, and allow them to immerse themselves in nature. It is a form of tourism that aligns with environmental and ecological projects [22] and can serve as a compelling case for sustainable and balanced regional development [23,24].

Another critical component of sustainable tourism is community involvement. Sustainable tourism encourages local communities to participate in decision-making and actively and ultimately benefit economically from their participation [10,25,26]. The involvement of communities is vital for fostering cohesion, improving outcomes, ensuring access, empowering communities, assisting local governments in making sustainable decisions, driving social transformation, and strengthening democracies [27]. Mureşan et al. (2016) emphasise that local communities view sustainable tourism as a positive force for their community by providing economic benefits and enhancing their quality of life [28]. Community-based tourism is owned and managed by the neighbourhood, for the neighbourhood, with the aim of enabling visitors to extend their awareness and learn about the community and local ways of life [29]. The strategy is advocated for its potential to address the social, environmental, and economic needs of local communities through tourism offerings [30] and it focuses more on equity of benefits through redistributive measures, while the profits directly benefit the local community [31].

Rural areas have often been characterised by their idyllic landscapes, unique culture, and tranquillity, but they are also known for their challenges, declining population, and economic stagnation. This situation is where sustainable tourism comes into play. It serves as a solution that will stimulate local economies and preserve cultural and natural heritage [32]. Rural development tries to revitalise these regions by offering visitors an authentic and immersive experience, such as eco-tourism. It wants to ensure that the traditional way of life stays intact while providing a tourist attraction [18]. Madeira is an exemplary model of successful rural tourism development through strategic diversification and dispersion of tourist activities. The island's deliberate investments in infrastructure, improved accessibility of rural areas, and a conscious shift in tourist distribution have resulted in above-average growth rates for the rural hinterland. The dispersion strategy mitigates the impact on urban infrastructure and has proven particularly effective under external pressures, where demand for quality rural spaces has increased [33]. Madeira's approach, incorporating policy measures and focusing on unique experiences like eco-tourism, showcases a proactive stance in shaping sustainable and diverse tourism, offering valuable lessons for regions navigating similar development trajectories.

*2.2. Strategic Documents in Croatia and EU Funding*

Developing green, resilient, and sustainable tourism is one of the defined goals of the acts on strategic planning in Croatia. The National Development Strategy until 2030 [34] is hierarchically the highest act of strategic planning that shapes and implements Croatia's development policies. In other words, the Strategy is a comprehensive document that aims to provide strategic guidance for all development policies and lower-level strategic planning documents, and it is in accordance with the Act on the System of Strategic Planning and Development Management of the Republic of Croatia [35]. The Ministry of Regional Development and EU Funds of the Republic of Croatia is defined as the coordinating body for the system of strategic planning and development management, meaning it organises and coordinates the process of preparation, implementation, monitoring of implementation, and reporting and evaluation on the implementation of the National Development Strategy until 2030 (NDS 2030). Four development directions are defined in the NDS 2030: 1. Sustainable economy and society, 2. Strengthening resistance to crises, 3. Green and digital transition, and 4. Balanced regional development. Four development directions define thirteen strategic goals [34]. Balanced regional development with its related strategic goals (12. Development of assisted areas and areas with developmental specialties and 13. Strengthening regional competitiveness) aligns with the case study presented in this paper.

The second document important for strategic planning in Croatia is the National Recovery and Resilience Plan 2021–2026 [36], which is the action plan of the Republic of Croatia in response to the crisis, through which the use of the EU funds from the Recovery and Resilience Facility is planned. The Recovery and Resilience Facility is a temporary instrument of the EU that provides grants and loans to support reforms and investments in the EU, and it is the centrepiece of NextGenerationEU. NextGenerationEU is the EU's instrument worth around 800 billion euros for supporting the economic recovery from the coronavirus pandemic and for building a greener, more digital, and more resilient future in the Member States between 2021 and 2026. Moreover, it aligns with the 2021–2027 Multiannual Financial Framework [37], which is the current EU long-term budget. To finance NextGenerationEU, the European Commission is, in the name of the EU Member States, borrowing money on the financial markets at more favourable rates and redistributing those amounts between the countries. To receive funds under the Recovery and Resilience Facility, Member States must prepare Recovery and Resilience Plans outlining how they plan to invest the funds. For that matter, Croatia adopted the National Recovery and Resilience Plan 2021–2026, which contains six components: (1) Economy, (2) Public administration, judiciary and state property, (3) Education, science and research, (4) Labour market and social protection, (5) Healthcare, and (6) Initiative: Renovation of buildings.

The last crucial strategic document for our article is the Development Plan of Dubrovnik-Neretva County until 2027, which states that there is an inequality in the level of tourism development among specific areas within the county, mostly between coastal parts and inland parts of the county [7]. Five Dubrovnik-Neretva county municipalities have been identified as assisted areas. Assisted areas are those of the Republic of Croatia that, based on the development index, have been evaluated as areas that lag behind the national average level of development and whose development should be further encouraged [38]. Most of these five municipalities belong to the inland areas of the county, and the main problems identified in these areas are emigration and an ageing population, as well as underdeveloped tourism and untapped natural potential [7].

## 3. Materials and Methods

The topic was designed as a case study by R.K. Yin, using document analysis and the cost–benefit analysis (CBA) method [39]. The analysis primarily relies on desk research as the data source method. Desk research is essential here as it provides a foundation of information, context, and benchmarks for a comprehensive and well-informed CBA. Within this framework, we employ a multifaceted approach to data estimation. Initially, our main data sources include local outlets such as municipalities and regional development agencies, in addition to publicly available data from various channels like the Croatian Bureau of Statistics. Unfortunately, interviews with local authorities and the population were not feasible due to the ongoing COVID-19 measures during the research period. To supplement our data, financial statements from analogous projects previously conducted by authors, executed in geographically similar yet distinct locations, serve as a foundational source of information, particularly those referring to returns. Concurrently, the estimation of costs hinges on official offers provided by suppliers for specific items. In one case where the official offer was not available, we conducted information through research within existing online shops that offer the needed items. In more detail, it refers to a digital column, which is planned to be set within the centre. This integration of the diverse data sources mentioned is the foundation for our comprehensive analysis.

The primary method used is cost–benefit analysis. This method is a crucial tool usually used to assess the feasibility and desirability of various public and private initiatives. In more detail, it is a systematic approach for evaluating a research idea's potential economic and social impacts, comparing its costs to the expected benefits. The origin of the CBA method is a topic of dispute. Some authors claim it emerged in the USA in the 1930s [40]. Others advocate that it emerged in the USA, but in 1808, and that it was developed by Albert Gallatin, who worked as the Secretary of Treasury [41], while the last theory states that French economist Jules Dupuit developed it in 1844 [42]. The first application of CBA in a research journal was in 1951, and since then, the use of the method has significantly increased, resulting in over 54 thousand publications by 2021 [43].

The method has wide applications in public infrastructure, such as roads, bridges, and airports; environmental policies, such as regulations to reduce pollution or protect endangered species; healthcare, for determining the efficiency of medical treatments and public health programmes; and educational projects, such as funding for schools or vocational training.

The main limitations of the method are that the CBA is (1) highly reliant on accurate data and assumptions, which can be challenging to obtain and are subject to errors and biases; (2) assigning monetary values to certain factors, like environmental impacts or human lives, which can be ethically and methodologically challenging; (3) it does not adequately account for risk and uncertainty, especially in long-term projects with uncertain future outcomes; (4) the method may not account for the distributional impacts of projects, potentially leading to inequitable outcomes; and (5) not all benefits can be easily monetised, such as improved quality of life or increased social cohesion, which may lead to an incomplete assessment.

Nevertheless, despite the limitations, the EU issued a CBA guide in 2014 [44]. It stated that the method is explicitly required, among other elements, as a basis for decision-making on the co-financing of major projects included in operational programs of the European Regional Development Fund (ERDF) and the Cohesion Fund (CF) to strengthen the economic, social, and territorial cohesion of the EU. Also, according to Dwyer (2012) and Burgan and Mules (2001), CBA is the most appropriate for most infrastructural projects, especially in regional and local tourism [45,46].

The standard CBA is carried out in seven steps: (1) Description of the context, (2) Definition of objectives, (3) Identification of the project, (4) Technical feasibility and environmental sustainability, (5) Financial analysis, (6) Economic analysis, and (7) Risk assessment [38] and we followed those steps in our analysis.

In more detail, the initial step included a meticulous exploration of several potential approaches to address the project's challenges effectively. Following the guidance provided by the European Commission [44], we considered three distinct options: (1) the do-nothing scenario, (2) the do-minimum scenario, and (3) the do-something-else scenario. This step served as the foundation for our subsequent evaluations.

The second phase of our analysis was the financial assessment, which strictly adhered to Annex III of European Commission Regulation 2015/207 (Annex III—Methodology for Cost–benefit Analysis) [47]. In this stage, we delved into the financial sustainability and profitability of the project by carefully scrutinising costs and revenues. The analysis used a 4% discount rate to evaluate the financial indicators.

Within financial analysis, the initial step encompassed estimating the magnitude and structure of costs. The implementation plans systematically organised these costs, encompassing distinct phases, including the pre-establishment phase, the development phase, and the ultimate assessment of financial sustainability. Our objective was to ascertain that the revenue generated by the centre's activities, such as workshops, rentals, and services, could adequately cover all anticipated costs. The core objective of this financial analysis was to gauge the return on investment and determine whether co-financing from EU funds was warranted. A critical criterion for EU funding eligibility was the financial net value of the investment, which, when we exclude contributions from external sources, must be negative. This condition indicated the need for non-repayable EU funds to execute the project effectively.

In this context, financial efficiency indicators, including the financial net present value (FNPV/C) and financial return rate on investment (FRR/C), were used to evaluate the project's viability without co-financing from EU funds. The FNPV/C is a comprehensive measure of financial investment sustainability, representing the sum of discounted net cash flows over the project's time horizon. The financial return rate on investment (FRR/C) delineates the discount rate at which the net financial value of the investment is zero. An important observation was that if the research idea failed to generate a financial profit before incorporating external financial support, it would be eligible for co-financing. Specifically, if the FNPV/C was less than 0 or the FRR/C was less than the applied discount rate, which stood at 4% in our analysis.

The third step was economic analysis, which assessed the project's economic justification by scrutinising its benefits and costs to society. This analysis considered both positive and negative impacts on society and the economy, considering multiplier effects, direct influences on the population, and social discount rates. Notably, the economic analysis also incorporated elements not considered in the financial analysis but held pivotal significance in the economic context. These adjustments entailed excluding value-added tax (VAT), social contributions, and other taxes or subsidies affecting prices, employing "shadow prices" to express the social opportunity costs of goods and services, and accounting for potential positive and negative impacts on the project that were beyond control. Furthermore, the recommended social discount rate of 5% was used to calculate the economic present value of the investment, representing a societal perspective on the valuation of future benefits and costs concerning the present.

To transform market prices into economic prices and account for market distortions, we applied conversion factors defined by EU-funded project standards. These factors played a pivotal role in monetizing the overall impact of the research idea. Notably, the economic analysis extended its scope by considering additional target groups that must be addressed in the financial analysis. Those groups consisted of the local population and entrepreneurs, who generated economic benefits by engaging in activities such as renting accommodation and providing hospitality services. Furthermore, the analysis acknowledged unquantifiable yet significant effects, including enhanced regional recognition, biodiversity preservation, and local partnership development.

The key economic indicators, economic net present value (ENPV), economic rate of return (ERR), and benefit–cost ratio (B/C), were used to assess the research idea's economic justification. ENPV was calculated as the disparity between the total discounted social benefits and costs. An ENPV greater than 0 signified that the benefits exceeded the costs, rendering the idea socially justified. ERR served as a measure of socio-economic profitability, with ERR surpassing the social discount rate deemed necessary for justifying the investment. Finally, the B/C ratio, comparing the net present value of benefits to the net present value of costs, needed to exceed 1 to establish the research idea as socially acceptable.

Sensitivity analysis was a critical component aimed at pinpointing the variables with the most significant impact on the research idea's financial efficiency and the indicators used in financial and economic analyses. Specifically, we evaluated three pivotal variables: (1) a 20% change in investment expenses, (2) a 20% change in operating revenues/economic benefits, and (3) a 20% change in investment and operational revenues/economic benefits. The primary objective was to identify the critical variables with the most pronounced influence on net present value (NPV) and internal rate of return (IRR) and gauge their effects on profitability.

The last stage of our CBA entailed a risk assessment. In this phase, we identified potential risks associated with the proposed project's implementation, assessed its likelihood impact, and outlined suitable management measures. The evaluation of risk likelihood and impact was conducted using a three-tiered scale: (1) denoted a low probability of occurrence and a minor impact, (2) indicated moderate probability of occurrence and a moderate impact, and (3) signified a high likelihood of occurrence and a significant impact.

## 4. Active Tourism Centre in Dubrovnik-Neretva County Inland Area

As evidenced by the literature review, Dubrovnik-Neretva County exhibits a significant disparity in its tourism development. The coastal areas, notably Dubrovnik, suffer from overcrowding, while the inland regions remain underdeveloped. Nonetheless, the untapped potential of the county's interior holds promises for sustainable tourism growth, offering numerous benefits to the local community. The benefits include the establishment of new lodging facilities, the stimulation of agriculture through creating new customer bases, the emergence of fresh dining options, and the introduction of supplementary amenities.

With this in mind, the primary objective of our case study was to promote the sustainable advancement of less-developed tourist destinations in Dubrovnik-Neretva County. Our example seeks to enhance residents' living standards and extend the tourist season. Furthermore, it aims to alleviate the strain on the heavily frequented coastal areas, such as the City of Dubrovnik, by generating new opportunities in the county's inland regions. These endeavours align seamlessly with the overarching goals of the EU's key policies, focusing on the green and digital transition. The target groups primarily included the local population and interested entrepreneurs who would ensure the implementation of the conceived content. In addition to them, the target groups also encompass active tourists, schools, children participating in nature school programs, tourist boards, relevant public authorities, and local government units.

The main activity of our study was the reconstruction and extension of the existing old building 'Vidonje' in Dubrovnik-Neretva county inland area into an active tourism centre.

The reconstruction of the old building has two parts. One part involves the construction of a basement-level building next to the building on the ground floor, which would include a water reservoir and some technical installations. The second part considers adding a floor within the building by creating a high attic on the first floor to minimise any alterations to the facade of the existing building, preserving its authenticity. During the design phase, careful consideration was given to ensuring uninterrupted occupancy and operation of the building while meeting all technical requirements in line with the principles of sustainable green and digital infrastructure. The building's power supply will be provided through alternative energy supply systems, specifically a photovoltaic power plant located on the building's roof. This system contributes to the increased use of renewable energy sources for heating, cooling, and electricity while significantly reducing air pollution. Furthermore, the building will have its own water reservoir as its primary water supply source. Heat pumps, another alternative solution, will use technical water. All of these measures categorise the future building as a nearly zero-energy building.

The purpose of the reconstructed and extended building is to create an active tourist facility where visitors and the local population can relax, rejuvenate, and engage in entertainment and education throughout the year. Following the active tourism concept, other activities include work on the surrounding forest, hiking and biking trails, equipment procurement, promotional activities, and project management activities.

One of the planned activities is bird-watching, for which the adaptation and renovation of a bird-watching facility are planned. Bird-watching involves various activities to observe and identify bird species in their natural habitats. Such tourism extends the tourist season, as it can be conducted pre-season and post-season. Additionally, indoors within the centre, a multimedia exhibit is planned to be installed, including a relief map of the birds and ecological materials, as well as an audio recording of bird songs that will provide an interpretation of the birdlife in the area. Those activities would be organized by local association experts specialising in nature protection from the Public Institution for Management of Protected Natural Areas of Dubrovnik-Neretva County.

## 5. Results

### 5.1. Options Analysis

The first step of the analysis involved considering various options to address the project challenges, namely (1) do nothing, (2) do minimum, and (3) do something else. The first option did not include additional activities and could have been more effective for tourism development, cost management, heritage preservation, or human capacity enhancement. The second option modified the project with limited investment, resulting in a modest positive impact. The third option introduced a new project concept, yielding slight positive effects but requiring moderate investment.

Comparing these options to the proposed research idea's effects, they offered fewer advantages. The proposed research idea provides a new attraction point, new accommodation infrastructure and overall restoration of the neglected facility. It also indirectly impacts private-sector tourism and establishes an infrastructure base for future tourism development. It strengthens the municipality's capacity for infrastructure investments and contributes to its overall tourism development. Therefore, the analysis shows that implementing the planned research idea and its associated activities is the best choice.

### 5.2. Financial Analysis

The second step of the CBA involved financial analysis, which spans 17 years, encompassing an investment phase of two years and a return phase of fifteen years, with costs related to project preparation included.

The financial analysis involved estimating and structuring costs according to the implementation plan. Operating revenue will come from service fees, rentals, guided tours, and space rentals. Prices are set for full cost recovery, with varying income assumptions for different services. Operational expenses include job creation, overhead, maintenance,

promotional and other costs over the reference period. The results show that generated revenues will only cover some costs, particularly in year ten, which requires significant investments. The remaining value of the centre after 17 years was considered, and the EU funding eligibility was assessed.

Financial efficiency indicators were used to determine the financial net value and return on investment. In the case without EU funding, the FNPV/C is negative and amounts to −1,244,319 euros, while the FRR/C is also negative at −14.5%. The negative values of both indicators show that the research idea could be more financially viable. In other words, it needs to generate more revenue over its lifespan to justify the financial investments required for its implementation, and the results confirm that non-repayable funds from an external source are necessary to carry out the research idea. The discounted net income is negative, totalling −642,404 euros for the reference period after the completion of the investment, meaning that there is no need to conduct a financial gap analysis and confirm that the idea of the centre is eligible for 100% financing.

The financial analysis revealed a negative cumulative net income for the research idea, indicating that operational revenues cannot cover expenses until 2040, resulting in a total deficit of −232,605 euros. The municipality must provide additional financial support to address cash flow gaps, especially in 2035. This supplementary funding from the local budget is easily manageable, as business revenues are more than enough to meet the centre's additional financial requirements until 2040. The results affirm the necessity of external funding, preferably from EU funds aligning with green and digital transition objectives, though it's acknowledged that short-term financial sustainability is feasible by incorporating revenue-generating activities; however, given the current opportunities and local context, a realistic perspective recognizes the need for external funding, with the anticipation that long-term financial sustainability will be achieved through additional activities in the future.

*5.3. Economic Analysis*

The economic analysis evaluated the societal and economic justification of the research idea, considering both benefits and costs. It examined the impact of investment on society and the economy, accounting for factors like taxes, shadow prices, and uncontrollable project-related impacts. The recommended social discount rate of 5% values future benefits and costs relative to the present. Conversion factors adjusted market prices to economic prices, reflecting market distortions. These factors are used for various cost categories in this research idea. The analysis aimed to monetise the overall impact by incorporating additional target groups, including the local population and entrepreneurs, whose economic benefits from activities such as renting accommodation and providing hospitality services were not initially considered in the financial analysis. Also, the project envisages activities related to the improvement of the offer for cyclists, that is, the establishment of information and communication technology digital solutions that will try to bring the wide range of active tourism in Dubrovnik-Neretva county closer to all potential 'outdoor' target groups. Except for cyclists, this also applies to mountaineers and all curious local and foreign visitors eager to explore nature and natural heritage. The analysis also recognised some significant effects that could not be quantified, such as enhancing the region's recognition, preserving biodiversity, and developing local partnerships. As of 2022, 1877 civil society organisations were officially registered within the Dubrovnik-Neretva County inland region. Many of these entities could serve as valuable partners in the project or stand as beneficiaries thereof.

Key economic indicators concluded that the research idea is socially justified: ENPV is positive, ERR is above the discount rate at 25.5%, and the B/C ratio is 3.67, indicating substantial economic benefits exceeding costs. The economic analysis supported the idea's funding from external sources.

### 5.4. Sensitivity Analysis and Risk Assessment

Sensitivity analysis identified all critical variables whose changes significantly impact the financial efficiency and the indicators used in financial and economic analysis. The variables used for this research idea were:

- 20% change in investment expenses;
- 20% change in operating revenues/economic benefits; and
- 20% change in investment and operational revenues/economic benefits.

The summary results of the sensitivity analysis are presented in Table 1.

**Table 1.** Change in parameters and their impact on profitability.

| Scenario | Change | FNPV | IRR | ENPV | ERR |
|---|---|---|---|---|---|
| Investment expenses | | | | | |
| Increase | +20% | −1,488,192 | −15.2% | 3,121,399 | 22.4% |
| Decrease | −20% | −1,041,091 | −13.7% | 3,416,995 | 28.7% |
| No change | 0 | −1,244,319 | −14.5% | 3,368,737 | 25.5% |
| Operational revenues/economic benefits | | | | | |
| Increase | +20% | −1,167,408 | −12.6% | 4,190,714 | 29.0% |
| Decrease | −20% | −1,308,411 | −16.1% | 2,532,573 | 22.1% |
| No change | 0 | −1,244,319 | −14.5% | 3,368,737 | 25.5% |
| Investment expenses and operational revenues/economic benefits | | | | | |
| Increase | +20% | −1,552,284 | −16.8% | 2,468,343 | 19.5% |
| Decrease | −20% | −964,180 | −11.7% | 4,451,917 | 32.8% |
| No change | 0% | −1,244,319 | −14.5% | 3,368,737 | 25.5% |

Source: authors calculations.

The sensitivity analysis revealed that changes in investment expenses have a similar effect on the research idea's economic viability as changes in operational revenues and economic benefits. The same holds for financial viability, particularly in scenarios involving increased expenses. Even with increased investment and operational expenses, the research idea remains justified regarding social benefits, showing a positive ENPV (+2,468,343 euros) and a significantly higher ERR (19.5% compared to 5%). While financial risks impact financial indicators negatively, the changes are relatively moderate.

The risk assessment showed (Table 2) that the highest risk for the research idea would be from delays in the execution of construction work and inflation. There is a significant likelihood that this risk will materialise because the construction market has not yet stabilised following the impact of the COVID-19 crisis and the war in Ukraine. Construction material prices in Croatia have risen significantly, and for some contractors, paying penalties for delays may be more cost-effective than continuing with construction. While the construction works are part of a well-developed plan with realistic timelines, the increase in market prices of certain goods can impact financial sustainability by raising costs in terms of investment and operations. Although this would significantly impact the research idea, the risk is of moderate probability as most prices have been pre-contracted and will not change.

**Table 2.** Description of risks and management measures.

| Risk Description | Risk Likelihood | Impact |
|---|---|---|
| Delays in the execution of construction works | High (3) | High (3) |
| Inflation | Moderate (2) | High (3) |
| Delay in the public procurement process | Low (1) | High (3) |
| Lack of interest from contractors to participate in public procurement | Low (1) | High (3) |
| Change in local political leadership | Low (1) | Moderate (2) |
| Failure to achieve projected revenues | Low (1) | Moderate (2) |
| Changes in team members | Low (1) | Low (1) |

Source: authors calculations.

## 6. Discussion

The case study on the creation of the active tourism centre in the Dubrovnik-Neretva County inland area contributes to the goals of the EU's key policies, focusing on the green and digital transition as well as the goals of the Croatian National Development Strategy until 2030 [34], as an essential document for ensuring sustainable advancement of less-developed tourist destinations. The research showcases a holistic approach by addressing multifaceted objectives, such as enhancing the quality of life for locals, minimising strain on densely populated tourist areas, and extending the tourism season. These objectives directly resonate with balancing tourism growth while preserving the environment and ensuring community well-being. By integrating renewable energy sources and water reservoirs for sustainable water supply and reconceptualising a neglected facility into a nearly zero-energy building, the project aligns with the European Green Deal [48], emphasising environmental conservation while offering sustainable tourism experiences. It also promotes sustainable rural development by introducing tourism featuring new eco-friendly and digitally enhanced outdoor activities. By fostering tourism in the inland areas, the county can expand its attractions, offering visitors a more diverse and varied set of expectations. The activities include hiking, bird-watching, and biking, which may not be easily accessible in urban settings.

The research activities, particularly the renovation of the bird-watching observatory in the ornithological reserve, are designed to foster and protect diverse bird species in their natural environments. This effort aligns with the European Green Deal, encouraging eco-friendly tourism activities that help conserve the local environment [49]. This focus on bird observation not only prolongs the tourism season following the Sustainable Tourism Development Strategy until 2030 [50], but also encourages tourists to discover the region's rich birdlife during off-peak seasons. Bird-watching, particularly in the active months of May and June, boosts the economy and raises environmental consciousness by attracting visitors keen on the region's preserved nature and diverse bird species. The case study also highlights the importance of inclusivity by providing educational and guided experiences for a broad audience, including children, adults, amateurs, and professionals. The project aims to offer a school-in-nature experience for local schools, providing children with an opportunity to learn about sustainability from the youngest age and connect with the environment. Educational activities thereby contribute to the region's long-term conservation and appreciation of its natural heritage.

The CBA method facilitated a structured and thorough analysis, comprehensively comparing the proposed plan and alternative options. By systematically weighing the costs and benefits of each activity, the CBA approach identified the unique advantages in favour of the selected research idea. This process aligned with the principles of sustainable tourism development and emphasised the initiative's ability to balance economic feasibility with positive environmental and societal impacts.

Recognising social opportunity costs, the expected influence on tourism growth, the subsequent rise in economic activities, and the involvement of local businesses and communities were essential. Furthermore, acknowledging immeasurable but significant advantages, such as heightened awareness of biodiversity and the development of local partnerships, amplifies the project's societal and environmental aspects. In the winter months, when the peak tourist season in Dubrovnik-Neretva County typically subsides from May to September, there is an opportunity to utilise the active tourist centre. During the off-season, it could be leased to the local mountaineering society, with the society covering the rent equivalent to monthly utilities. Such collaboration would contribute to the centre's financial sustainability and support the local mountaineering society's role in enhancing community bonds, promoting health and fitness, fostering environmental stewardship, and facilitating educational and skill development opportunities. Furthermore, the analysis acknowledged unquantifiable yet significant effects, like the instilment of a sense of pride in the community, enhancing its image and attractiveness. By renting the centre to the mountaineering society, the centre can mitigate the impact of seasonality and provide a consistent source of revenue and activity for the community. However, a potential constraint is the difficulty in accurately quantifying all elements, particularly non-market values. Assessing impacts like the fortification of local partnerships could be subjective and complex to measure precisely, potentially leading to underestimating these contributions.

The risk assessment conducted as part of the CBA showcased a meticulous approach to identifying and mitigating potential challenges. It scrutinised various risks, including market volatility and implementation-related challenges, that could impede the initiative's success. Proactive measures were identified as crucial safeguards to counteract market fluctuations and challenges during implementation, ensuring the initiative's readiness to navigate uncertainties and maintain momentum towards sustainable development within the local area.

The local initiatives align harmoniously with broader European strategies, particularly the European Green Deal [48]. The eco-tourism initiative in Dubrovnik-Neretva County mirrors the aspirations of European policies by embracing climate change mitigation strategies and promoting resource-efficient building and renovation while exemplifying a commitment to a greener economy and sustainable tourism. By aligning with this EU policy, the region aims to comply with environmental conservation practices and to drive a more cohesive, sustainable, and cooperative approach towards tourism and environmental preservation. These efforts benefit the country and contribute to the EU's broader ambitions for a resource-efficient and environmentally sustainable economy. The mutual alignment between local and regional initiatives and EU policies underscores a shared commitment to sustainability and environmental conservation, paving the way for a more concerted and practical approach to fostering a harmonious and sustainable tourism landscape [51,52].

## 7. Conclusions

In conclusion, this research significantly contributes to understanding and promoting sustainable tourism in Dubrovnik-Neretva County. Focusing on tourism development in inland areas addresses the challenges posed by overcrowded tourist hotspots and lays the groundwork for more diversified and sustainable tourism. Developing rural areas and their tourism is essential when trying to decrease the overcrowding of popular urban cities. It helps preserve cultural and natural heritage, prevents further environmental degradation, and makes tourism more sustainable and longer-lasting, which in turn helps the local communities and fosters their economic benefits.

The analysis of potential options led to the conclusion that none of the alternative approaches offers the numerous advantages presented by the proposed project, which introduces new attractions, enhances accommodation infrastructure, restores neglected facilities, and has significant indirect impacts on private-sector tourism. Regarding Hypothesis 1 (H1), financial investments in Dubrovnik-Neretva inland areas encourage touristic

development. The results showed that proposed investments establish an infrastructure base for future tourism development, strengthen the municipality's capacity for infrastructure investments, and contribute to overall tourism development in the municipality. As such, the results provide support for Hypothesis 1 (H1) of our research. Therefore, the optimal choice is to proceed with the planned project and its associated activities. Regarding Hypothesis 2 (H2), revitalization of less-developed areas requires external funding. The financial analysis indicated that the proposed idea needs to generate more revenue to cover costs. As a result, external funding is necessary for the project's viability. The cumulative net income remains negative until 2040, with the municipality providing supplementary financial support to bridge the cash flow gap. Therefore, the results from our research provide support for Hypothesis 2 (H2).

The economic analysis showed that the proposed initiative is socially justified, with positive economic indicators such as ENPV, ERR, and B/C ratio. Sensitivity analysis highlighted the impact of various parameter changes on profitability, demonstrating the proposals' robustness even in scenarios involving increased expenses. The final section of the CBA assessed risks related to the research idea's implementation, identifying potential risks and their likelihood and impact, which, in conclusion, can be managed effectively.

The principal constraints of the study primarily stem from the need for more connectivity between the tourism centre and other forthcoming initiatives in the adjacent municipalities. The project's financial worth and potential impact on the reduction of tourist numbers in Dubrovnik could be significantly enhanced if additional municipalities were to collaborate. Another limitation is that the CBA is executed at a specific point in time. Given the prevailing inflationary conditions within the European Union, future price uncertainties may arise. Consequently, there may be a necessity for adjustments to the CBA to ensure its continued accuracy and relevance. The last identified limitation is relying on historical data to project future outcomes. The analysis heavily depends on past trends and patterns, assuming that future conditions will follow a similar trajectory. However, unforeseen economic, social, or environmental changes could render the projections unreliable. This limitation highlights the need for a more dynamic and adaptive approach that considers the potential for unpredictable events or shifts in circumstances that may affect the outcomes of the proposed tourism project.

Therefore, future studies might include the opinions and attitudes of the local population or propose models for collaborative tourism development involving multiple municipalities with the aim of a more sustainable reduction in tourist numbers in popular destinations like Dubrovnik.

In conclusion, the active tourism centre in Dubrovnik-Neretva County's inland area demonstrated how cost–benefit analysis can contribute to investments in tourist infrastructure in less-developed rural areas. The project offered a viable solution through eco-friendly outdoor activities, with the recommendation of securing external funding for its successful implementation. It is expected that these investments encourage socio-economical and environmentally sustainable local and regional development.

**Author Contributions:** Conceptualisation, M.F. and I.B.; methodology, M.F. and I.B.; writing—original draft preparation, M.F., I.B. and A.L.S.; writing—review and editing, M.F. and I.B.; visualisation, I.B.; supervision, M.F.; project administration, I.B.; funding acquisition: M.F. All authors have read and agreed to the published version of the manuscript.

**Funding:** This research was funded by the Municipality of Zažablje, a contract for the procurement of services for the preparation of a Feasibility Study with a cost–benefit analysis for the construction of the Centre for Active Tourism of the Neretva Valley; class: 334-01/22-01/01, registry number: 2117-22-02-22-6.

**Institutional Review Board Statement:** Not applicable.

**Informed Consent Statement:** Not applicable.

**Data Availability Statement:** No new data were created or analyzed in this study. Data sharing is not applicable to this article.

**Conflicts of Interest:** The authors declare no conflict of interest. The funders had no role in the design of the study; in the collection, analyses, or interpretation of data; in the writing of the manuscript, or in the decision to publish the results.

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
