# Peer review of "Revitalizing Rural Tourism: A Croatian Case Study in Sustainable Practices"

_sustainability, doi:10.3390/su16010031_

Round 1

Reviewer 1 Report

Comments and Suggestions for Authors

The aim of the paper is clear. The abstract is descriptive enough and manage to incorporate the description of the entire article.

The authors must explicitly present their hypotheses. According to Robert K. Yin, who was cited as a source for the research method chosen in this article, he emphasized in another paper that case studies should include elements such as research questions and hypotheses, a delineation of the research design, apparatus, and data collection procedures, the presentation of collected data, data analysis, and a discussion of findings and conclusions.

One of the key elements of sustainable tourism, community engagement/involvement and local population, are mentioned many times in literature review and this is a positive aspect. I rather suggest adding more references regarding this very important topic. And I also recommend incorporating discussions on community-based tourism, a strategy advocated for its potential to address the social, environmental, and economic needs of local communities through tourism offerings. Moreover, it is strongly advised to enhance the level of detail in the Results and Discussion section, specifically elucidating the methods and outcomes of community engagement in the presented case study. This would provide a more comprehensive understanding of the practical application of community involvement in the context of the study.

The conclusions, limitations, and future research lack depth from diverse perspectives, and they are not adequately contextualized to generate fresh avenues for future studies. The authors should offer a more comprehensive exploration of the study's limitations or clarify if they assert the absence of limitations and elucidate the rationale behind such a claim.

Reviewer 2 Report

Comments and Suggestions for Authors

First, I would like to thank the authors for their case study based research. Some observations are provided below for better understanding.

1) In the introduction section, this study just discussed the need for expansion of the Dubrovnik-Neretva County inland. Why these expansions are needed and what is the current situation of Dubrovnik-Neretva County inland, should be included in the introduction section, so that the audience can understand better the importance of the study.

2) Please provide a sub-title for the first section of the literature review. Also please highlight the vital problems of Dubrovnik-Neretva County inland that need to be addressed.

3) Please justify the need for Desk Research. 

4) "We conducted information through research within existing online web shops that offer the needed items". What is the need to manage data from online web shops? Please justify. 

5) "This integration of diverse data sources is the foundation for our comprehensive analysis". Please mention the names of diverse data sources.

6) Please check the format of the in-text citation. Such as y R.K. Yin [33].

Reviewer 3 Report

Comments and Suggestions for Authors

Great study; hence, more enhancements to several areas of the study will greatly increase its potentiality.

- Financial Viability and Revenue Generation: The project's financial analysis reveals that, in the absence of EU funding, it would not be economically sustainable. The project must generate additional revenue throughout its duration to justify the necessary financial investments. The negative Financial Net Present Value (FNPV/C) and Financial Return Rate on Investment (FRR/C) indicate the need for non-repayable funds from external sources. Hence, conducting a more comprehensive analysis of methods to generate revenue could enhance the findings of the study.

- The project is in line with the EU's major policies that prioritize green and digital transitions, as well as the Croatian National Development Strategy. Nevertheless, there is a possibility to incorporate these more extensive strategies into the project's structure in order to guarantee complete synchronization. Hence, investigating ways to reinforce the link between the local initiative and wider policies will improve the significance and efficacy of the study.

- Community Involvement and Local Partnerships: Although the study acknowledges the involvement of local populations and entrepreneurs, a more thorough examination of active engagement and collaboration with community groups and local organizations could amplify the study's influence and reception. This may entail the implementation of community-led initiatives or the collaborative development of tourism activities that accurately showcase the local culture and heritage.

By addressing these gaps and implementing the suggested improvements, the study could enhance its effectiveness, sustainability, and alignment with broader strategic goals, thereby contributing more significantly to the development of sustainable tourism in Dubrovnik-Neretva County.

Round 2

Reviewer 1 Report

Comments and Suggestions for Authors

The authors have effectively addressed all the concerns raised during the initial review, resulting in a substantial improvement. The paper is now significantly improved and deserves to be published in its current form.

Reviewer 3 Report

Comments and Suggestions for Authors

I believe the authors managed the reviewer's comments and improved the manuscript. Great overall additions.